# Phytochemical Profiling of *Coryphantha macromeris* (Cactaceae) Growing in Greenhouse Conditions Using Ultra-High-Performance Liquid Chromatography–Tandem Mass Spectrometry

**DOI:** 10.3390/molecules24040705

**Published:** 2019-02-15

**Authors:** Emmanuel Cabañas-García, Carlos Areche, Juan Jáuregui-Rincón, Francisco Cruz-Sosa, Eugenio Pérez-Molphe Balch

**Affiliations:** 1Centro de Ciencias Básicas, Universidad Autónoma de Aguascalientes, Av. Universidad 940, 20131 Aguascalientes, Mexico; fde_garci@hotmail.com (E.C.-G.); jjaureg@correo.uaa.mx (J.J.-R.); eperezmb@correo.uaa.mx (E.P.-M.B.); 2Departamento de Química, Facultad de Ciencias, Universidad de Chile, Casilla 653, Santiago 7800024, Chile; areche@uchile.cl; 3Departamento de Biotecnología, Universidad Autónoma Metropolitana-Iztapalapa. Av. San Rafael Atlixco 186, Col. Vicentina C.P., 09340 Ciudad de México, Mexico

**Keywords:** Cactaceae, phenolic compounds, secondary metabolites, UHPLC, succulent plants, fragmentation pattern, active compounds

## Abstract

Chromatographic separation combined with mass spectrometry is a powerful tool for the characterization of plant metabolites because of its high sensitivity and selectivity. In this work, the phytochemical profile of aerial and radicular parts of *Coryphantha macromeris* (Engelm.) Britton & Rose growing under greenhouse conditions was qualitatively investigated for the first time by means of modern ultra-high-performance liquid chromatography–tandem mass spectrometry (UHPLC-PDA-HESI-Orbitrap-MS/MS). The UHPLC-PDA-HESI-Orbitrap-MS/MS analysis indicated a high complexity in phenolic metabolites. In our investigation, 69 compounds were detected and 60 of them were identified. Among detected compounds, several phenolic acids, phenolic glycosides, and organic acids were found. Within this diversity, 26 metabolites were exclusively detected in the aerial part, and 19 in the roots. Twenty-four metabolites occurred in both plant parts. According to the relative abundance of peaks in the chromatogram, ferulic and piscidic acids and their derivatives may correspond to one of the main phenolic compounds of *C. macromeris*. Our results contribute to the phytochemical knowledge regarding *C. macromeris* and its potential applications in the pharmaceutical and cosmetic industries. Besides, some metabolites and their fragmentation patterns are reported here for the first time for cacti species.

## 1. Introduction

The family Cactaceae is one of the most threatened within the plant kingdom [1], and is the most important plant family of the arid and semiarid regions of America [2], comprising around 1600 species [3]. Cacti are succulent plants which have been used for their functional properties and industrial utility [4,5,6]. Different cacti species obtained by cultivated or wild-collected methods have been used as food sources and fodder, for ornamental purposes, and for medicinal purposes (due to their antioxidant [7], antimutagenic [8], and bactericidal [9] properties), and have been cataloged as a source of bioactive compounds including alkaloids [10,11] and phenolics [12,13].

Plants contain different classes of metabolites found in different concentrations and of a wide structural diversity. Their phytochemical profiles and functional properties may vary according to: a) the solvent and extraction processes [14,15], b) the stage of growth [16], c) storage conditions [17], and d) the specific section of the plant [18,19]. This complexity makes it difficult to identify plant metabolites through a single analytical method or under a defined separation system [20]. Several techniques have been proposed to achieve the elucidation of the plant metabolome, including the coupling of mass spectrometry with different separation techniques [21,22]. The knowledge of natural product chemistry is important since identified metabolites may serve as the basis for the discovery or design of novel compounds with biological activities [23] and as chemotaxonomic markers for plant classification [24].

The genus *Coryphantha* is highly endemic, distributed in the arid and semiarid regions located between the Sierra Madre Oriental and the Sierra Madre Occidental in northern Mexico, expanding to the southern regions of Arizona, Texas, and New Mexico, USA [25]. Currently, natural populations of *Coryphantha* species are reduced due to overexploitation and destruction of habitats. *Coryphantha* spp. are slow growing plants and the environmental conditions of semiarid regions make their germination, growth, and reproduction difficult; for this reason, biotechnological approaches have been proposed for the *in vitro* propagation of *Coryphantha* spp. and other cacti species [26,27]. *Coryphantha macromeris* (Engelm.) Britton & Rose is a globose cactus traditionally known as “Dona Ana”, “long mamma cory-cactus”, “big needle cactus”, and “biznaga partida-partida”; it is mainly distributed in northern Mexico and southern United States. In Mexico, *C. macromeris* is used in folk medicine for healing stomach disorders; nevertheless, hallucinogenic properties have been proposed, due to the presence of alkaloids such as macromerine [28] and other β-phenethylamine derivatives [29]. More recently, Kikuchi, et al. [30] evaluated the chemical constituents of a psychotropic herbal product and, using DNA sequence analysis, suggested that *C. macromeris* could be one of its components, as the *rpl16* intron sequence of the herbal product showed 99% of similarity with that of *C. macromeris.* Information regarding the phytochemical composition and bioactive compounds in *C. macromeris* is scarce. Thus, the aim of this study was to identify, using UHPLC coupled with tandem mass spectrometry, the main secondary metabolites present in shoots and roots of *Coryphantha macromeris* growing under greenhouse conditions in order to contribute to the phytochemical knowledge of this species and its potential applications.

## 2. Results and Discussion

Methanolic extracts prepared with aerial and radicular parts of *C. macromeris* cultivated under greenhouse conditions were analyzed by ultra-high-performance liquid chromatography–tandem mass spectrometry and a photodiode array detector (UHPLC-PDA-HESI-Orbitrap-MS/MS). This technique has been used for the characterization of metabolites in complex mixtures [31], and the chromatographic conditions used in this work have shown efficiency for the separation and identification of different secondary metabolites [32,33,34]. For *C. macromeris* extracts, UHPLC conditions allowed a good separation of many sample components.

The phytochemical characterization of *C. macromeris* was achieved by comparing the obtained information by UHPLC-PDA-HESI-Orbitrap-MS/MS with spectroscopic evidences existing in the literature or by structure searching and studying the fragmentation pattern of molecules. Additionally, retention time and UV (λ_max_) spectra were used for peak characterization. As far as we know, metabolite profiling of *C. macromeris* is reported for the first time in this work. Elution profile (Figure 1) and mass spectra obtained in negative ion mode indicated a highly complex phenolic composition. Peak characteristics (i.e., retention time, theoretical and measured mass, UV (λ_max_), and fragmentation pattern) and tentative identification of each compound are summarized in Table 1. All detected compounds exhibited an accuracy smaller than 5 ppm. Under the proposed UHPLC-PDA-HESI-Orbitrap-MS/MS method, 69 compounds were separated, and 60 of them were tentatively identified in *C. macromeris* greenhouse plants (Table 1). Within this complexity, 26 compounds occurred in the aerial part, 19 in occurred in the roots, and 24 occurred in both plant parts, suggesting a translocation mechanism from actively photosynthetic tissues to metabolite consuming/storage areas.

Compounds **1**–**5** were detected in both parts of *C. macromeris*. Compound **1** was assigned as vaccihein A (Figure 2a). This antioxidant metabolite was isolated for the first time in *Vaccinium ashei* (Ericaceae) by Ono, et al. [35] and this is the first time that is reported for cacti species. Compounds **2**–**5** were assigned as carboxylic acids: compounds **2** and **3** were proposed as dihydroxy methoxy butanoic acid and malic acid [36], respectively, and compounds **4** and **5** as isocitric acid isomers [13]. Citric acid occurs as one of the main hydrophilic constituents in *Opuntia* species (Cactaceae) [13,37], and its presence may be also related to the crassulacean acid metabolism of cacti species [12].

Among the detected compounds, 31 metabolites were phenolic acids (compounds **6**–**13**, **15**–**25**, **31**, **32**, **34**, **36**–**42**, **44**, **48** and **62**) and 16 of them occurred in the aerial part (see Table 1). In our investigation, compound **6** was assigned as 3,5-dihydroxy-4-methyloxolan-2-yl methoxy-6-hydroxymethyl oxane-3,4,5-triol, due to the presence of fragments ions at *m/z*: 293.12424, 279.10870, 147.06578, and 131.07106 (see Table 1); Vankudothu and Anwar [38] proposed, through molecular docking, that this metabolite showed high affinity for the ACC2 protein, suggesting an effect against type 2 diabetes. Thus, the elicitation and isolation of this compound may be a potential use for *C. macromeris*. Also, two protocatechuic acid derivatives (compounds **9** and **16**) were detected: compound **9** was proposed as protocatechuic acid hexoside (see Figure 2b) and compound **16** as protocatechuic aldehyde since pseudomolecular ion at *m/z*: 137.02386 yielded fragments at *m/z*: 121.02882 and 109.02884 (see Table 1). It has been proposed that protocatechuic aldehyde influences the pharmacokinetic activity of medicinal herbal extracts prepared with *Salvia miltiorrhiza* (Lamiaceae) [39], suggesting that the presence of this compound in *C. macromeris* extract may enhance its functional properties. Protocatechuic acid has been found as an aglycone in other cacti species such as *Opuntia ficus-indica* [40] and *Myrtillocactus geometrizans* [17], and as one of the main constituents in *Opuntia humifusa* [18].

Additionally, compound **13** was only detected in the aerial part and was assigned as lucuminic acid, since the pseudomolecular ion yielded fragments at *m/z*: 163.03947, 119.04939 and 107.04942 (see Table 1). Lucuminic acid has been reported for *Calocarpum sapota* (Sapotaceae) by Takeda, et al. [41] and this is the first time that is reported for cacti species. In our investigation, sinapic acid (compound **20**) and two of its derivatives (compounds **19** and **25**) were also detected and characterized as reported previously [42]. Sinapic acid is synthesized from ferulic acid through enzymatic mechanisms [43] and it has been proposed to have anti-inflammatory activities [44].

In the same way, other phenolic metabolites such as two syringic acid derivatives (compounds **21** and **23**), two caffeic acid isomers [45] (compounds **24** and **45**), and cinnamic and gallic acid derivatives (compound **31** and **59**, respectively) were also found (see Table 1). In addition, piscidic and ferulic acid and/or their derivatives were detected as the most recurrent phenolic acids in *C. macromeris* extracts: six isomers of piscidic acid (compounds **7**, **8**, **10**, **11**, **12**, and **15**) and four if its derivatives (compounds **17**, **18**, **22**, and **37**) were detected and assigned as reported previously [46,47]. Piscidic acid has been previously identified in juices prepared with *Opuntia ficus-indica* fruits [13] and according to the number of peaks detected and the relative abundance shown in the chromatogram, piscidic acid may be one of the main constituents of the *C. macromeris* aerial part (see Figure 1a, Table 1); further studies are required to confirm the structure of this metabolite and its total concentration in each part of the plant. On the other hand, compounds **28**, **33**, **34**, and **36** were assigned as ferulic acid isomers [48], and compounds **29**, **35**, **38**, **39**, and **41** as ferulic acid derivatives, since they showed characteristic fragment ions of ferulic acid (see Table 1). Our results indicate that ferulic acid is present in both parts of the plant but with a higher relative abundance in the shoots (Figure 1a). Ferulic acid has been detected in different cacti species [18,37,49,50] and its antiproliferative effect in human colon carcinoma HT29 cell line [51] as well its therapeutic potential against cardiovascular and neurodegenerative disorders [52] has been proposed. Our results suggest that *C. macromeris* could represent a potential source for the isolation of this compound or its derivatives; further studies are required to isolate and characterize the structure of these compounds (isomers) and to know their concentration, elicitation, and production in biotechnological systems.

In addition to compounds **2**–**5**, compounds **26**, **27**, **32**, **40**, **44**, **47**, **51**–**53**, **55**, and **63** were also assigned as carboxylic acids. The fragmentation pattern of these metabolites was mainly characterized by the loss of water (see Table 1). Among these metabolites, compounds **51** and **55** were detected in both parts of the plant and assigned as corchorifatty acid F isomers. This metabolite has been previously identified in leaves of *Corchorus olitorius* (Tiliaceae) by Yoshikawa, et al. [53] and in aerial parts of *Chaenomeles sinensis* (Rosaceae) [54]. It also confers activity against pathogenic fungus *Pyricularia oryzae* in resistant rice cultivars [55]. To our knowledge, this is the first time that is reported for cacti species. On the other hand, compound **40** was identified as azelaic acid, according to Liu, et al. [56]. This dicarboxylic acid has been used for the treatment of acne and skin disorders [57,58] and skin hyperpigmentation [59]. Compound **52** was assigned as tianshic acid due to the presence of fragments at *m/z*: 165.12788 and 127.11205 (see Table 1). The occurrence of this metabolite has been previously reported for *Sambucus williamsii* (Adoxaceae) by Yang, et al. [60] and no information exists for cacti species.

In the same way, other polar compounds were characterized in *C. macromeris* methanolic extracts. Compound **14** was assigned as hyrtioerectine C (alkaloid). Youssef [61] isolated this alkaloid and proposed its cytotoxic activity against HeLa cells. In our research group, studies are being carried out to elucidate the presence of other alkaloids in, and the potential applications of, *C. macromeris* extracts. Signals revealed for compound **30** indicated the presence of a glucopyranoside derivative [62] in the aerial part of *C. macromeris*. Similarly, compound **42** was tentatively identified as 2-phenylethyl β-D-glucopyranoside. This phenolic glycoside has been previously reported in *Pachysandra terminalis* (Buxaceae) [63] and in *Lactuca indica* (Asteraceae) [64]. To our knowledge, this is the first time that it is reported for cacti species. Compound **43** was tentatively identified as dalbergioidin (see Figure 2d); this antifungal isoflavone was reported in *Vigna angularis* (Fabaceae) as one of the mechanisms against *Phytophthora vignae* zoospores germination [65], suggesting that its presence in the *C. macromeris* aerial part may correspond to a constitutive adaptation mechanism since no pathogenic conditions were given to the plants; further studies are required to assess this hypothesis. Compounds **46**, **48**, and **50** were also assigned as metabolites of glycosidic nature (see Table 1). On the other hand, the pseudomolecular ion and fragmentation pattern of compound **49** indicated the presence of buteine (chalcone; see Figure 2e) in the radicular part of *C. macromeris*. Buteine exists in *Rhus verniciflua* (Anacardiaceae), a traditional herb used for cancer treatment [66], and its antioxidant activity [67] and potential as a chemotherapeutic agent [68] has been proposed. Compounds **57** and **60** were tentatively identified as nordihidrocapsiate isomers (see Figure 2f) and compound **58** as plastoquinone 3.

Finally, compounds **54**, **56**, **61**, **62**, and **65**–**69** were not identified since molecular information obtained did not match with theoretical information existing in the literature. Absorption spectrum of these compounds suggest that they may contain benzoic acid within its structure (UV λ_max_: 280 nm). Further studies are required for structural elucidation of these metabolites.

## 3. Materials and Methods

### 3.1. Plant Material

The plants of *Coryphantha macromeris* were obtained from the *in vitro* germplasm bank of the Autonomous University of Aguascalientes, México. *Coryphantha macromeris* was maintained on semisolid Murashige and Skoog medium [69] (3% sucrose, 8 g·L^−1^ agar, and pH 5.7) and incubated at 25 °C with fluorescent light (40 μmol·m^2^·s^−1^) and 16/8 (light/dark) photoperiod. Each plant was subcultured for three months during one year with the described conditions, and then *in vitro* plants were acclimated to greenhouse conditions as reported previously [26] and kept for one year until harvesting for phytochemical analysis. The plant material was botanically identified by Professor Miguel Alvarado Rodríguez. A voucher specimen was deposited at the herbarium of the Autonomous University of Aguascalientes (HUAA; Voucher No. 6386).

### 3.2. Sample Preparation

For analysis, one-year-old plants growing under greenhouse conditions were collected and separated into aerial and radicular parts. Each part of the plant was sliced and dried in an oven (40 °C) during 1 week in dark conditions. Dried material was finally pulverized in a mortar and then extracted three times with methanol in an ultrasonic bath (30 min each time). The resultant extract was filtered and evaporated under reduced pressure at 40 °C and freeze-dried (Labconco Freeze Dryer; Labconco Corporation, Kansas City, MO, USA). Each freeze-dried sample was resuspended (2.5 mg·mL^−1^) in HPLC-MS-grade methanol and sonicated over 10 min. All samples were filtered (0.22 μm) and injected in an UHPLC system hyphenated with a mass spectrometer, as given below.

### 3.3. UHPLC-PDA-HESI-Orbitrap-MS/MS Conditions

Phytochemical analysis was performed as reported previously [32,33,34], using a Dionex Ultimate 3000 UHPLC system (Thermo Fisher Scientific, Bremen, Germany) with a C18 column (ID: 150 × 4.6 mm, 5 μm; Restek Corporation, Bellefonte, PA, USA) and equipped with a Quaternary Series RS pump and a Dionex Ultimate 3000 Series TCC-3000RS column compartments with an Ultimate 3000 Series WPS-3000RS autosampler (Thermo Fisher Scientific) and a rapid separations PDA detector. The detection wavelengths were 254, 280, 320, and 440 nm, and PDA was recorded from 200 to 800 nm for peak characterization. The separation was performed in a gradient elution mode composed by 1% formic aqueous solution (A) and acetonitrile (B). The flow rate was 1.0 mL·min^−1^ and the injection volume 10 μL. The gradient program [time (min), %B] was: (0.00, 5), (5.00, 5), (10.00, 30), (15.00, 30), (20.00, 70), (25.00, 70), (35.00, 5), and 12 min for column equilibration before each injection. The system was controlled by Chromeleon 7.2 Software (Thermo Fisher Scientific, Waltham, MA, USA and Dionex Softron GmbH division of Thermo Fisher Scientific) and hyphenated with a Thermo high resolution Q Exactive focus mass spectrometer (Thermo Fisher Scientific). The chromatographic system was coupled to the mass spectrometer with a heated electrospray ionization source II (HESI II). Nitrogen (purity > 99.999%) was employed as both the collision and damping gas. Nitrogen was obtained from a Genius NM32LA nitrogen generator (Peak Scientific, Billerica, MA, USA). Mass calibration for Orbitrap was performed once a week, in both negative and positive modes. Caffeine and *N*-butylamine (Sigma-Aldrich, Saint Louis, MO, USA) were the calibration standards for positive ions and buspirone hydrochloride, sodium dodecyl sulfate, and taurocholic acid sodium salt were used to calibrate the mass spectrometer. These compounds were dissolved in a mixture of acetic acid, acetonitrile, water, and methanol (Merck Darmstadt, Hesse, Germany) and infused using a Chemyx Fusion 100 syringe pump. XCalibur 2.3 software and Trace Finder 3.2 (Thermo Fisher Scientific, San Jose, CA, USA) were used for UHPLC control and data processing, respectively. Q Exactive 2.0 SP 2 (Thermo Fisher Scientific, Waltham, MA, USA) was used to control the mass spectrometer. 

#### MS Parameters

The HESI parameters were optimized as follows: sheath gas flow rate 75 units; auxiliary gas flow rate 20 units; capillary temperature 400 °C; auxiliary gas heater temperature 500 °C; spray voltage 2500 V (for ESI-); and S lens RF level 30. Full scan data in negative mode was acquired at a resolving power of 70,000 full width half maximum (FWHM) at *m*/*z* 200. For the compounds of interest, a scan range of *m*/*z* 100–1000 was chosen; the automatic gain control (AGC) was set at 3 × 10^6^ and the injection time was set to 200 ms. Scan rate was set at 2 scans s^−1^. External calibration was performed using a calibration solution in positive and negative modes before each sample series. In addition to the full scan acquisition method, for confirmation purposes, a targeted MS/MS analysis was performed using the mass inclusion list and expected retention times of the target analytes, with a 30 s time window, with the Orbitrap spectrometer operating both in positive and negative mode at 17,500 FWHM (*m*/*z* 200). The AGC target was set to 2 × 10^5^, with the maximum injection time of 20 ms. The precursor ions were filtered by the quadrupole operating at an isolation window of *m*/*z* 2. The fore vacuum, high vacuum, and ultra-high vacuum were maintained at approximately 2 mbar, from 105 to below 1010 mbar, respectively. Collision energy (HCD cell) was operated at 30 eV. Detection was based on calculated exact mass and on retention time of target compounds presented in Table 1. The mass tolerance window was set to 5 ppm.

## 4. Conclusions

The utilized procedures allowed the separation of 69 metabolites present in methanolic extracts of aerial and radicular parts of *C. macromeris,* and the identification of most of them. The aerial part showed higher diversity of metabolites compared with roots, and 24 metabolites occurred in both plant parts. Among detected compounds, ferulic and piscidic acid and their derivatives were the most recurrent phenolic metabolites in *C. macromeris*. In addition, different classes of compounds with reported functional properties were detected. To our knowledge, the phytochemical profile of aerial and root parts of *C. macromeris* is reported here for the first time by means of modern ultra-high-performance liquid chromatography–tandem mass spectrometry. Based on the detected compounds, *C. macromeris* may find potential applications in the pharmaceutical and cosmetic industries. The obtained information could be also useful as chemotaxonomic markers for *Coryphantha* species.

## Figures and Tables

**Figure 1 molecules-24-00705-f001:**
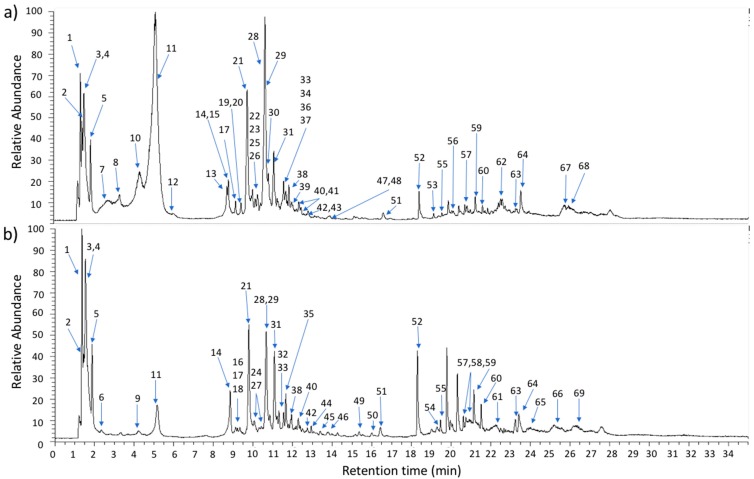
Ultra-high-performance liquid chromatography (UHPLC) chromatogram of *Coryphantha macromeris* methanolic extracts prepared with aerial (**a**) and radicular (**b**) parts. Peak numbers refer to the metabolites indicated in Table 1; repeated numbers in (**a**) or (**b**) indicate that the metabolite is found in both plant parts.

**Figure 2 molecules-24-00705-f002:**
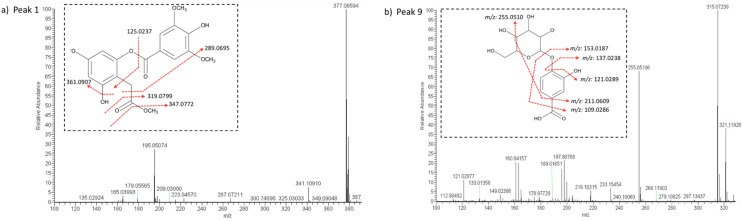
Full scan of some metabolites identified for the first time in *C. macromeris*. Dotted inset represents the molecule and red dotted lines in each inset represent the proposed fragmentation pattern. Peak numbers in each figure, refer to the compounds indicated in Table 1.

**Table 1 molecules-24-00705-t001:** Metabolites identified in aerial and radicular parts of *Coryphantha macromeris* by ultra-high-performance liquid chromatography–tandem mass spectrometry (UHPLC-PDA-HESI-Orbitrap-MS/MS) data using heated electrospray ionization source (HESI) in negative ion mode.

Peak	Retention Time (min)	$\mathrm{UV}\;\mathrm{Max}\;(\lambda_{\max})$	Tentative Identification	Elemental Composition [M-H]^−^	Theoretical Mass (*m*/*z*)	Measured Mass (*m*/*z*)	Accuracy (ppm)	MSn Ions	Plant Part
1	1.37	220, 272	Vaccihein A	C_18_H_17_O_9_^−^	377.08781	377.08600	4.80	361.09076 ([M-H-OH]^−^)	Root/shoot
								347.07724 ([M-H-OCH_3_]^−^)	
								319.07990 ([M-H-C_2_H_3_O_2_]^−^)	
								289.06958 ([M-H-C_2_H_3_O_2_-OCH_3_]^−^)	
								125.02375 ([M-H-C_8_H_9_O_4_-C_3_H_5_O_2_]^−^)	
2	1.45	220, 272	Dihydroxy methoxy butanoic acid	C_5_H_9_O_5_^−^	149.04555	149.04498	3.82	135.02925 ([M-H-CH_3_]^−^)	Root/shoot
								131.03433 ([M-H-OH]^−^)	
								119.03425 ([M-H-CH_3_-OH]^−^)	
								103.03932 ([M-H-OCH_3_-OH]^−^)	
3	1.52	220, 274	2-Hydroxy-succinic acid (malic acid)	C_4_H_5_O_5_^−^	133.01390	133.01363	2.03	115.00297 ([M-H-OH]^−^)	Root/shoot
4	1.55	224, 277	Iso citric acid	C_6_H_7_O_7_^−^	191.01973	191.01938	1.83	111.00790 ([M-H-CO_2_-2OH]^−^)	Root/shoot
5	1.87	277	Iso citric acid isomer	C_6_H_7_O_7_^−^	191.01973	191.01938	1.83	111.00790 ([M-H-CO_2_-2OH]^−^)	Root/shoot
6	2.35	220, 277	3,5-Dihydroxy-4-methyloxolan-2-yl methoxy-6-hydroxymethyl oxane-3,4,5-triol	C_12_H_21_O_9_^−^	309.11911	309.11935	0.78	293.12424 ([M-H-OH]^−^)	Root
								279.10870 ([M-H-OH-CH_3_]^−^)	
								147.06578 ([M-H-C_6_H_11_O_5_]^−^)	
								131.07106 ([M-H-C_6_H_11_O_5_-OH]^−^)	
7	2.74	224, 277	Piscidic acid isomer	C_11_H_11_O_7_^−^	255.05103	255.05098	0.20	193.05013 ([M-H-CHO_2_-OH]^−^)	Shoot
								165.05521 ([M-H-C_2_H_2_O_3_-OH]^−^)	
								135.04442 ([M-H-C_2_H_2_O_3_-CHO_2_]^−^)	
								119.04952 ([M-H-C_2_H_2_O_3_-CH_2_-OH]^−^)	
								107.04927 ([M-H-C_4_H_4_O_3_]^−^)	
8	3.34	228, 277	Piscidic acid isomer	C_11_H_11_O_7_^−^	255.05103	255.05098	0.20	193.05013 ([M-H-CHO_2_-OH]^−^)	Shoot
								165.05521 ([M-H-C_2_H_2_O_3_-OH]^−^)	
								135.04442 ([M-H-C_2_H_2_O_3_-CHO_2_]^−^)	
								119.04952 ([M-H-C_2_H_2_O_3_-CH_2_-OH]^−^)	
								107.04927 ([M-H-C_4_H_4_O_3_]^−^)	
9	4.22	197, 223, 278	Protocatechuic acid hesoxide	C_13_H_15_O_9_^−^	315.07230	315.07239	0.29	255.05104 ([M-H-C_2_H_5_O_2_]^−^)	Root
								211.06094 ([M-H-C_2_H_5_O_2_-CHO_2_]^−^)	
								153.01878 ([M-H-C_6_H_11_O_5_]^−^)	
								137.02385 ([M-H-C_6_H_11_O_5_-OH]^−^)	
								121.02895 ([M-H-C_6_H_11_O_5_-2OH]^−^)	
								109.02863 ([M-H-C_6_H_11_O_5_-CHO_2_]^−^)	
10	4.34	223, 276	Piscidic acid isomer	C_11_H_11_O_7_^−^	255.05103	255.05098	0.20	193.05013 ([M-H-CHO_2_-OH]^−^)	Shoot
								165.05521([M-H-C_2_H_2_O_3_-OH]^−^)	
								135.04442 ([M-H-C_2_H_2_O_3_-CHO_2_]^−^)	
								119.04952 ([M-H-C_2_H_2_O_3_-CH_2_-OH]^−^)	
								107.04927 ([M-H-C_4_H_4_O_3_]^−^)	
11	5.13	222, 275	Piscidic acid isomer	C_11_H_11_O_7_^−^	255.05103	255.05095	0.31	193.05013 ([M-H-CHO_2_-OH]^−^)	Root/shoot
								165.05521 ([M-H-C_2_H_2_O_3_-OH]^−^)	
								135.04442 ([M-H-C_2_H_2_O_3_-CHO_2_]^−^)	
								119.04952 ([M-H-C_2_H_2_O_3_-CH_2_-OH]^−^)	
								107.04927 ([M-H-C_4_H_4_O_3_]^−^)	
12	6.02	222, 275	Piscidic acid isomer	C_11_H_11_O_7_^−^	255.05103	255.05101	0.08	193.05013 ([M-H-CHO_2_-OH]^−^)	Shoot
								165.05521([M-H-C_2_H_2_O_3_-OH]^−^)	
								135.04442 ([M-H-C_2_H_2_O_3_-CHO_2_]^−^)	
								119.04952 ([M-H-C_2_H_2_O_3_-CH_2_-OH]^−^)	
								107.04927 ([M-H-C_4_H_4_O_3_]^−^)	
13	8.74	222, 277	Lucuminic acid	C_19_H_25_O_12_^−^	445.13515	445.13531	0.36	163.03947 ([M-H-C_10_H_19_O_8_-OH]^−^)	Shoot
								119.04939 ([M-H-C_11_H_19_O_10_-OH]^−^)	
								107.04942 ([M-H-C_11_H_19_O_9_-CHO_2_]^−^)	
14	8.85	223, 276	Hyrtioerectine C	C_11_H_12_NO_4_^−^	222.07718	222.07703	0.68	206.08206 ([M-H-OH]^−^)	Root/shoot
								198.07718 ([M-H-C_2_H_3_]^−^)	
								180.06580 ([M-H-C_2_H_3_-OH]^−^)	
								178.08685 ([M-H-CHO_2_]^−^)	
15	8.93	227, 283	Piscidic acid isomer	C_11_H_11_O_7_^−^	255.05103	255.05104	0.04	193.05013 ([M-H-CHO_2_-OH]^−^)	Shoot
								165.05521([M-H-C_2_H_2_O_3_-OH]^−^)	
								135.04442 ([M-H-C_2_H_2_O_3_-CHO_2_]^−^)	
								119.04952 ([M-H-C_2_H_2_O_3_-CH_2_-OH]^−^)	
								107.04927 ([M-H-C_4_H_4_O_3_]^−^)	
16	9.12	255, 207	Protocatechuic aldehyde	C_7_H_5_O_3_^−^	137.02442	137.02386	4.09	121.02882 ([M-H-OH]^−^)	Root
								109.02884 ([M-H-COH]^−^)	
17	9.19	230, 286	Piscidic acid derivative	C_21_H_27_O_13_^−^	-	487.14600	-	255.05110 ([piscidic acid]^−^)	Root/shoot
								193.05078 ([piscidic acid-CHO_2_-OH]^−^)	
								165.05516 ([piscidic acid-C_2_H_2_O_3_-OH]^−^	
								135.04453 ([piscidic acid-C_2_H_2_O_3_-CHO_2_]^−^) 107.04935 ([piscidic acid-C_4_H_4_O_3_]^−^)	
18	9.41	223, 278	Piscidic acid derivative	C_20_H_27_O_13_^−^	-	475.14606	-	255.05112 ([piscidic acid]^−^)	Root
								193.05037 ([piscidic acid-CHO_2_-OH]^−^)	
								165.05513([piscidic acid-C_2_H_2_O_3_-OH]^−^)	
								135.04453 ([piscidic acid-C_2_H_2_O_3_-CHO_2_]^−^)	
								107.04923 ([piscidic acid-C_4_H_4_O_3_]^−^)	
19	9.45	231, 295	Sinapic acid derivative	C_22_H_29_O_14_^−^	-	517.15649	-	223.06104 ([sinapic acid]^−^)	Shoot
								208.03767 ([sinapic acid-CH_3_]^−^)	
								179.07083 ([sinapic acid- CHO_2_]^−^)	
								164.04738 ([sinapic acid-CHO_2_-OH]^−^)	
20	9.50	231, 295	Sinapic acid	C_11_H_11_O_5_^−^	223.06070	223.06102	1.43	208.03757 ([M-2H-CH_3_]^−^)	Shoot
								179.07094 ([M-H-CHO_2_]^−^)	
								164.04730 ([M-2H-CHO_2_-OH]^−^)	
21	9.77	224, 276	Syringic acid acetate	C_11_H_11_O_6_^−^	239.05611	239.05594	0.71	197.04517 ([syringic acid]^−^)	Root/shoot
								195.06580 ([M-H-CHO_2_]^−^)	
								179.03439 ([M-H-2CH_3_O]^−^)	
								149.06023 [M-H-2CH_3_O-OH]^−^)	
								135.04456 ([M-H-CHO_2_-2CH_3_O]^−^)	
								107.04944 ([M-H-CHO_2_-CH_3_O-C_2_H_3_O_2_]^−^)	
22	9.97	231, 286	Piscidic acid derivative	C_21_H_27_O_13_^−^	-	487.14603	-	255.05101 ([piscidic acid]^−^)	Shoot
								193.05025 ([piscidic acid-CHO_2_-OH]^−^)	
								165.05528 ([piscidic acid-C_2_H_2_O_3_-OH]^−^)	
								135.04456 ([piscidic acid-C_2_H_2_O_3_-CHO_2_]^−^)	
								107.04945 ([piscidic acid-C_4_H_4_O_3_]^−^)	
23	10.04	233, 283	Syringic acid acetate derivative	C_19_H_31_O_8_^−^	-	387.20276	-	239.05594 ([syringic acid acetate]^−^)	Shoot
								197.04517 ([syringic acid]^−^)	
								195.06580 ([syringic acid acetate-CHO_2_]^−^)	
								179.03439 ([syringic acid acetate-2CH_3_O]^−^)	
								149.06023 ([syringic acid acetate-2CH_3_O-OH]^−^)	
								135.04456 ([syringic acid acetate-CHO_2_-2CH_3_O]^−^)	
								107.04944 ([syringic acid acetate-CHO_2_-CH_3_O-C_2_H_3_O_2_]^−^)	
24	10.08	231, 283	Caffeic acid	C_9_H_7_O_4_^−^	179.03498	179.03477	1.17	163.03950 ([M-H-OH]^−^)	Root
								135.04510 ([M-H-CHO_2_]^−^)	
								109.02870 ([M-H-C_3_H_3_O_2_]^−^)	
25	10.17	231, 286	Sinapic acid hexoside	C_17_H_21_O_10_^−^	385.11402	385.11438	0.93	223.06099 ([M-H-C_6_H_11_O_5_]^−^)	Shoot
								208.03757 ([sinapic acid-CH_3_]^−^)	
								179.07095 ([sinapic acid-CHO_2_]^−^)	
								164.04745 ([sinapic acid-CHO_2_-OH]^−^)	
26	10.27	234, 283	Propanedioic acid, [5-[[2-[(6-deoxy- α-l-galactopyranosyl) oxy] cyclohexyl] oxy]-Pentyl]	C_20_H_33_O_10_^−^	433.20792	433.20825	0.76	417.21347 ([M-H-OH]^−^)	Shoot
								387.20309 ([M-H-CHO_2_]^−^)	
								287.15030 ([M-H-C_6_H_11_O_4_]^−^)	
								245.13950 ([M-H-C_8_H_13_O_5_]^−^)	
								131.07069 ([M-H-C_14_H_23_O_6_]^−^)	
27	10.31	237, 283	Cyclohexanecarboxylic acid, 3-[(6-deoxy-3-*O*-methyl-d-galactopyranosyl)oxy]-1,4,5-trihydroxy	C_14_H_23_O_10_^−^	351.12967	351.13010	1.22	303.14487 ([M-H-3OH]^−^)	Root
								287.14999 ([M-H-4OH]^−^)	
								273.13449 ([M-H-3OH-CH_3_O]^−^)	
28	10.63	235, 326	Ferulic acid	C_10_H_9_O_4_^−^	193.05063	193.05032	1.61	179.03455 ([M-H-CH_3_]^−^)	Root/shoot
								149.06050 ([M-H-CHO_2_]^−^)	
								163.03963 ([M-H-CH_3_-OH]^−^)	
								147.04456 ([M-H-CH_3_-2OH]^−^)	
29	10.69	235, 327	Ferulic acid derivative (fertaric acid)	C_14_H_13_O_9_^−^	325.05651	325.05664	0.40	193.05032 ([ferulic acid]^−^)	Root/shoot
								179.03453 ([ferulic acid-CH_3_]^−^)	
								163.03954 ([ferulic acid-CH_3_-OH]^−^)	
30	10.84	237, 291	7,8,11-Trihydroxyguai-4-en- 3-one 8-*O*-β-d-glucopyranoside	C_21_H_33_O_9_^−^	429.21250	429.21335	1.98	267.16003 ([M-H-C_6_H_11_O_5_]^−^)	Shoot
								249.14989 ([M-H-C_6_H_11_O_5_-OH]^−^)	
31	11.09	235, 286	Cinnamic acid derivative	C_8_H_14_O_6_^−^	-	206.08205	-	147.04449 ([cinnamic acid]^−^)	Root/shoot
								103.05447 ([cinnamic acid-CHO_2_]^−^)	
32	11.22	227, 283	2-Propenoic acid, 2-methyl-, 4-[2-(2,4-dioxo-1,5-dioxaspiro [5.5]undec-3-yl)ethenyl]-6-(2,4-dioxo-1,5-dioxaspiro[5.5]undec-3-ylidene)-4-hexenyl ester	C_30_H_34_O_10_^−^	554.21629	554.21448	3.27	193.05049 ([M-H-C_20_H_25_O_6_]^−^)	Root
33	11.54	239, 289, 323	Ferulic acid isomer	C_10_H_9_O_4_^−^	193.05063	193.05048	0.78	179.03458 ([M-H-CH_3_]^−^)	Root/shoot
								149.06030 ([M-H-CHO_2_]^−^)	
								163.03954 ([M-H-CH_3_-OH]^−^)	
34	11.61	238, 292, 323	Ferulic acid isomer	C_10_H_9_O_4_^−^	193.05063	193.05038	1.29	179.03441 ([M-H-CH_3_]^−^)	Shoot
								149.06026 ([M-H-CHO_2_]^−^)	
								163.03937 ([M-H-CH_3_-OH]^−^)	
								147.04457 ([M-H-CH_3_-2OH]^−^)	
35	11.66	226, 282	Ferulic acid derivative I	C_24_H_24_O_5_^−^	-	392.16193	-	193.05038 ([ferulic acid]^−^)	Root
								149.06030 ([ferulic acid-CHO_2_]^−^)	
								163.03954 ([ferulic acid-CH_3_-OH]^−^)	
36	11.71	236, 286	Ferulic acid isomer	C_10_H_9_O_4_^−^	193.05063	193.05038	1.29	179.03452 ([M-H-CH_3_]^−^)	Shoot
								149.06024 ([M-H-CHO_2_]^−^)	
								163.03937 ([M-H-CH_3_-OH]^−^)	
								147.04440 ([M-H-CH_3_-2OH]^−^)	
37	11.79	235, 286, 380	2-Isoferulic piscidic acid-1-metyl ester	C_22_H_21_O_10_^−^	445.11400	445.11438	0.85	255.05092 ([piscidic acid]^−^)	Shoot
								193.050380 ([ferulic acid]^−^)	
								165.05516 ([piscidic acid-C_2_H_2_O_3_-OH]^−^)	
								135.04440 ([piscidic acid-C_2_H_2_O_3_-CHO_2_]^−^)	
								107.04936 ([piscidic acid-C_4_H_4_O_3_]^−^)	
38	11.86	240, 296, 381	Ferulic acid derivative II	C_20_H_29_O_10_^−^	-	429.17707	-	193.05029 ([ferulic acid]^−^)	Root/shoot
								179.03450 ([ferulic acid-CH_3_]^−^)	
								163.03937 ([ferulic acid-CH_3_-OH]^−^)	
								147.04440 ([ferulic acid-CH_3_-2OH]^−^)	
39	12.18	283, 368	Ferulic acid derivative III	C_21_H_31_O_13_^−^	-	491.17731	-	193.05026 ([ferulic acid]^−^)	Shoot
								179.03456 ([ferulic acid-CH_3_]^−^)	
								163.03929 ([ferulic acid-CH_3_-OH]^−^)	
								147.04446 ([ferulic acid-CH_3_-2OH]^−^)	
40	12.38	283	Azelaic acid	C_9_H_15_O_4_^−^	187.09758	187.09740	0.96	169.08130 ([M-H-OH]^−^)	Root/shoot
								125.09650 ([M-H-CHO_2_-OH]^−^)	
41	12.47	284, 368	Ferulic acid derivative IV	C_20_H_29_O_10_^−^	-	429.17706	-	193.05023 ([ferulic acid]^−^)	Shoot
								179.03427 ([ferulic acid-CH_3_]^−^)	
								163.03958 ([ferulic acid-CH_3_-OH]^−^)	
								147.04450 ([ferulic acid-CH_3_-2OH]^−^)	
42	12.70	259	2-Phenylethyl β-d-glucopyranoside	C_14_H_19_O_6_^−^	283.11871	283.11893	0.78	267.12402 ([M-H-OH]^−^)	Root/shoot
								251.12881([M-H-2OH]^−^)	
								235.13390 ([M-H-3OH]^−^)	
								121.06506([M-H-C_6_H_11_O_5_]^−^)	
43	12.75	272, 368	Dalbergioidin	C_15_H_11_O_6_^−^	287.05611	287.05618	0.24	271.06094 ([M-H-OH]^−^)	Shoot
								179.03467 ([M-H-C_6_H_5_O_2_]^−^)	
								165.05252 ([M-H-C_6_H_5_O_3_]^−^)	
								163.03951 ([M-H-C_6_H_5_O_2_-OH]^−^)	
								147.04404 ([M-H-C_6_H_5_O_2_-2OH]^−^)	
								125.02380 ([M-H-C_9_H_9_O_3_]^−^)	
								109.02868 ([M-H-C_9_H_7_O_4_]^−^)	
44	12.95	260	4,8,12-trihydroxy-2,4-dodecadienoic acid,	C_12_H_19_O_5_^−^	243.1238	243.12383	0.12	199.1337 ([M-H-CHO_2_]^−^)	Root
								139.11221 ([M-H-CHO_2_-C_2_H_3_-2OH]^−^)	
45	13.40	283	Caffeic acid isomer	C_9_H_7_O_4_^−^	179.03498	179.03481	0.95	163.03954 ([M-H-OH]^−^)	Root
								135.04454 ([M-H-CHO_2_]^−^)	
								109.02881([M-H-C_3_H_3_O_2_]^−^)	
46	13.80	222, 284	β-d-Glucopyranoside, 1,1-dimethyl-5-methylenenonyl	C_10_H_17_O_4_^−^	345.22826	345.22849	0.67	327.21780 ([M-H-OH]^−^)	Root
								315.21780 ([M-H-CH_3_O]^−^)	
47	13.90	224, 284	Sebacic acid	C_10_H_17_O_4_^−^	201.11323	201.11293	1.49	185.11778 ([M-H-OH]^−^)	Shoot
								157.12276 ([M-H-CHO_2_]^−^)	
48	13.93	223, 284	alpha-Ionol O-[arabinosyl-(1->6)-glucoside]	C_24_H_39_O_10_^−^	487.25487	487.25504	0.35	473.24008 ([M-H-CH_3_]^−^)	Shoot
								459.22311 ([M-H-2CH_3_]^−^)	
								355.21292 ([M-H-C_5_H_9_O_4_]^−^)	
								341.19687 ([M-H-C_5_H_9_O_4_-CH_3_]^−^)	
49	15.35	283, 368,	Buteine	C_15_H_11_O_5_^−^	271.0612	271.06131	0.41	163.03952 ([M-H-C_6_H_5_O_2_]^−^)	Root
								137.02380 ([M-H-C_8_H_7_O_2_]^−^	
								135.04443 ([M-H-C_7_H_5_O_3_]^−^)	
								121.02880 ([M-H-C_8_H_7_O_2_-OH]^−^)	
								108.02104 ([M-H-C_9_H_7_O_3_]^−^)	
50	15.98	283	D-xylofuranose tetradecyl glycoside	C_22_H_41_O_9_^−^	449.27561	449.27576	0.33	403.27036 ([M-H-CH_3_O-OH]^−^)	Root
								316.22061 ([M-H-C_5_H_9_O_4_]^−^)	
								329.23349 ([M-H-C_4_H_9_-4OH]^−^)	
								117.05499 ([M-H-C_17_H_32_O_5_-OH]^−^)	
51	16.42	283	Corchorifatty acid F isomer	C_18_H_31_O_5_^−^	327.21770	327.21799	0.89	309.20665 ([M-H-OH]^−^)	Root/shoot
								291.19684 ([M-H-2OH]^−^)	
								173.11787 ([M-H-C_9_H_15_O_2_]^−^)	
								157.12346 ([M-H-C_9_H_15_O_2_-OH]^−^)	
								125.09643([M-H-C_3_H_5_O_2_-C_7_H_13_O_2_]^−^)	
52	18.30	283, 368	Tianshic acid	C_18_H_33_O_5_^−^	329.23335	329.23361	0.79	165.12788 ([M-H-C_7_H_15_O-3OH]^−^)	Root/shoot
								127.11205 ([M-H-C_10_H_19_O_2_-2OH]^−^)	
53	19.15	283	Dimethyl sebacate (sebacic acid derivative)	C_12_H_21_O_4_^−^	229.14453	229.144	0.39	201.11287 ([sebacic acid]^−^)	Shoot
								215.12865 ([M-H-CH_3_]^−^)	
								211.13374 ([M-H-O]^−^)	
								199.13379 ([M-H-CH_3_O]^−^)	
								185.11778 ([M-H-CH_3_O-CH_3_]^−^)	
								157.12303 ([M-H-C_2_H_3_O_2_-CH_3_]^−^)	
54	19.28	282, 368	Unknown	C_13_H_27_O_8_^−^	-	311.16888	-	-	Root
55	19.47	283, 368	Corchorifatty acid F isomer	C_18_H_31_O_5_^−^	327.21770	327.21802	0.98	309.20731 ([M-H-OH]^−^)	Root/shoot
								291.19672 ([M-H-2OH]^−^)	
								173.11792 ([M-H-C_9_H_15_O_2_]^−^)	
								125.09679 ([M-H-C_3_H_5_O_2_-C_7_H_13_O_2_]^−^)	
56	20.06	283	Unknown	C_13_H_27_O_8_^−^	-	311.16888		-	Shoot
57	20.63	283	Nordihydrocapsiate	C_17_H_25_O_4_^−^	293.17583	293.17612	0.99	277.18088 ([M-H-OH]^−^)	Root/shoot
								263.16534 ([M-H-CH_3_O]^−^)	
								247.16968 ([M-H-CH_3_O-OH]^−^)	
								157.12309 ([M-H-C_8_H_9_O_2_]^−^)	
								153.05524 ([M-H-C_9_H_17_O]^−^)	
								141.12810 ([M-H-C_8_H_9_O_3_]^−^)	
58	20.92	274	Plastoquinone 3	C_23_H_31_O_2_^−^	339.23296	339.23322	0.77	203.10753 ([M-H-C_9_H_15_]^−^)	Root
								163.11229 ([M-H-C_6_H_11_-C_5_H_7_O]^−^)	
								149.06009 ([M-H-C_13_H_21_]^−^)	
								135.04454 ([M-H-C_14_H_23_]^−^)	
59	21.25	283	Decyl gallate (gallic acid derivative)	C_17_H_25_O_5_^−^	309.17075	309.17093	0.58	293.17935 ([M-H-OH]^−^)	Root/shoot
								169.01381 ([gallic acid]^−^)	
								153.01903([M-H-C_10_H_21_O]^−^)	
								125.02367 ([M-H-C_11_H_21_O_2_]^−^)	
60	21.61	283	Nordihydrocapsiate isomer	C_17_H_25_O_4_^−^	293.17583	293.17612	0.99	277.18080 ([M-H-OH]^−^)	Root/shoot
								263.16535 ([M-H-CH_3_O]^−^)	
								247.16990 ([M-H-CH_3_O-OH]^−^)	
								157.12309 ([M-H-C_8_H_9_O_2_]^−^)	
								153.05524 ([M-H-C_9_H_17_O]^−^)	
								141.12810 ([M-H-C_8_H_9_O_3_]^−^)	
61	22.30	283	Unknown	C_13_H_27_O_8_^−^	-	311.16904	-	-	Root
62	22.53	283	Unknown	C_24_H_45_O_11_^−^	-	509.29691	-	-	Shoot
63	23.36	283, 337	13-Hydroxyoctadecadienoic acid	C_18_H_31_O_3_^−^	295.22787	295.22797	0.34	281.21204 ([M-H-CH_3_]^−^)	Root/shoot
								279.23334 ([M-H-OH]^−^)	
								169.12331 ([M-H-C_8_H_15_O]^−^)	
								153.12767 ([M-H-C_8_H_15_O-OH]^−^)	
64	23.54	283, 337	p-Hydroxynonanophenone	C_15_H_21_O_2_^−^	233.1547	233.15462	0.34	219.17544 ([M-H-O]^−^)	Root/shoot
								167.14342 ([M-H-C_4_H_5_O]^−^)	
								135.04446 ([M-H-C_7_H_15_]^−^)	
								121.02875 ([M-H-C_7_H_15_-CH_3_]^−^)	
65	24.03	283	Unknown	C_15_H_31_O_8_^−^	-	339.20029	-	-	Root
66	25.20	283	Unknown	C_14_H_29_O_8_^−^	-	325.18463	-	-	Root
67	25.72	283, 337	Unknown	C_14_H_29_O_8_^−^	-	325.18457	-	-	Shoot
68	25.98	283, 337	Unknown	C_14_H_29_O_8_^−^	-	325.18454	-	-	Shoot
69	26.24	283	Unknown	C_14_H_29_O_8_^−^	-	325.18463	-	-	Root

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
