# Peer review of "Phytochemical Profiling of Coryphantha macromeris (Cactaceae) Growing in Greenhouse Conditions Using Ultra-High-Performance Liquid Chromatography–Tandem Mass Spectrometry"

_molecules, 2019, doi:10.3390/molecules24040705_

Round 1

Reviewer 1 Report

Manuscript describes the chemical composition of a succulent plant, Coryphanta macromeris, endemic to Mexico and Arizona. The phytochemical profile has been analysed by the advanced UHPLC-PDA-HESI-Orbitrap-MS/MS method. In general manuscript is well organised.

Specific comments:

- A moderate English revision is needed (see for example line 40. 56-57, etc).

- I would suggest not to use "section"; use instead "plant organs", "plant parts", or similar expressions

- Line 44: instead of "maturation", use "stage of growth".

- Plant stage of growth should also be indicated in MM. 

- In MM Authors should also describe when they collected the roots, at what stage of growth of the plant?

-Figure 2: I would suggest to indicate directly on the figures the number of compounds (1, 9, etc..) 

- I wonder why a quantitative determination  has not be made. I would suggest to carry it out. 

- In RD,  comparative comments of analytical data from other species of Coryphanta or similar cacti should be added.

Author Response

Attached comments Reviewer 1

Reviewer 2 Report

This manuscript investigated phytochemical compounds in the aerial and root parts of planted Coryphantha macromeris (Engelm.) Britton & Rose by LC-Orbitrap-MS/MS. A total of 60 compounds were assigned. It could be published on Molecules.
All the compounds were only assigned by UV and MS, therefore, all the identifications should be changed to “temporarily identified” throughout the Results and Discussion.

Author Response

Attached comments Reviewer 2
